# Potemkin Understanding in Large Language Models

**Marina Mancoridis** [1]   **Bec Weeks** [2]   **Keyon Vafa** [3]   **Sendhil Mullainathan** [4]

## Abstract

Large language models (LLMs) are regularly evaluated using benchmark datasets. But what justifies making inferences about an LLM's capabilities based on its answers to a curated set of questions? This paper first introduces a formal framework to address this question. The key is to note that the benchmarks used to test LLMs—such as AP exams—are also those used to test people. However, this raises an implication: such benchmarks are only valid tests if LLMs misunderstand concepts in ways that mirror human misunderstandings. Otherwise, success on benchmarks only demonstrates **potemkin understanding:** the illusion of understanding driven by answers irreconcilable with how any human would interpret a concept. We present two procedures for quantifying the existence of potemkins: one using a specially designed benchmark in three domains, the other using a general procedure that provides a lower-bound on their prevalence. We find that potemkins are ubiquitous across models, tasks, and domains. We also find that these failures reflect not just incorrect understanding, but deeper internal incoherence in concept representations.

## 1. Introduction

There has been a marked change in how we interpret machine learning evaluations. Today, large language models (LLMs) are evaluated on benchmark datasets: curated questions with rubrics for grading. Success on these questions is treated as evidence of broader conceptual understanding. Previously, while benchmarks were also used to evaluate supervised learning models, success was interpreted more narrowly. A pathology classifier that performs well on X-ray classification is not credited with an understanding of

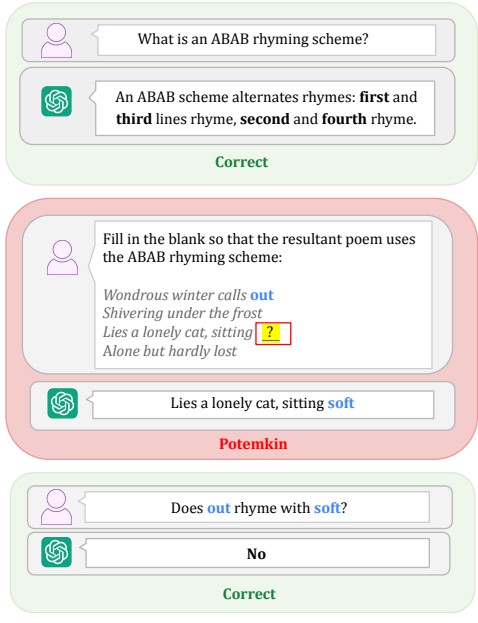

Figure 1. *Illustration of potemkin understanding in a large language model*. This example shows GPT-4o's failure to apply its own conceptual explanation of an ABAB rhyming scheme.

vision—we only draw inferences for its performance on specific distributions, inferences that are limited by distribution shift. What justifies interpreting benchmark performance this way for LLMs?

This paper first introduces a framework that justifies using benchmarks to measure concept understanding in LLMs. The central insight is simple: we already use analogous tests to assess humans. AP exams, AIME math competitions, and coding challenges are credible measures precisely because humans cannot excel on these benchmarks without a true grasp of the required concepts. This is because the ways in which a human might misunderstand a concept are small and structured; there is no human who can do well on an AP math exam without understanding basic math concepts.

This framework also raises an implication: that human benchmarks are only valid tests for LLMs if LLMs misunderstand concepts in the same way that humans do. If LLM misunderstandings diverge from human patterns, models can succeed on benchmarks without truly understanding

---

[1]Massachusetts Institute of Technology [2]University of Chicago [3]Harvard University [4]Massachusetts Institute of Technology. Correspondence to: Marina Mancoridis <marinam@mit.edu>.

*Proceedings of the 42nd International Conference on Machine Learning*, Vancouver, Canada. PMLR 267, 2025. Copyright 2025 by the author(s).

the underlying concepts. When this happens, it results in pathologies we call **potemkins**[1].

Figure 1 illustrates a potemkin. When an LLM is asked to explain an ABAB rhyming scheme, its response is clear and correct (top panel). At first glance, it may appear that the LLM has understood the concept, in the same way that a human with the provided explanation would understand. However, when tasked to generate text in an ABAB rhyming scheme, the LLM fails, producing non-rhyming words (middle panel). Strikingly, the LLM seems to recognize that its output does not rhyme (bottom panel). This specific combination of correct and incorrect answers is irreconcilable with any answer that a human would give.

Potemkins occur when an LLM performs well on tasks that would indicate conceptual understanding if a human completed them, but do not indicate understanding in the LLM. This paper develops two procedures for measuring the prevalence of potemkins in LLMs. The first is tailored to a specific kind of potemkin: the divide between an LLM's ability to explain a concept and apply it. We collect a benchmark dataset across three domains — literary techniques, game theory, and psychological biases — designed to measure the prevalence of this type of potemkins. In contrast, the second procedure is general and doesn't make assumptions on the structure or domain of potemkins, but it only provides a lower-bound on their prevalence.

We apply these procedures to a set of LLMs and find that potemkins are ubiquitous. For example, despite models being able to define concepts in each domain in our benchmark dataset near-perfectly, they struggle to apply these concepts accurately. We find that potemkins are not arising due merely to incorrect understanding of concepts, but rather due to incoherence. Despite the fact that the automated procedure provides only a lower bound, it still identifies high rates of potemkins across LLMs on questions from common benchmark datasets.

The rest of the paper is structured as follows: Section 2 presents our framework for defining potemkins. Section 3 introduces a benchmark designed to measure the presence of potemkins and shows their ubiquity across key domains. Section 4 corroborates these findings using a separate, automated evaluation procedure.

## 2. Framework

Potemkins arise when there is a misalignment between how humans and large language models (LLMs) understand concepts. Here, we present a theoretical framework for defining conceptual understanding.

---

[1]This term comes from Potemkin villages—elaborate facades built to create the illusion of substance where none actually exists.

**Conceptual understanding in people.** At a high level, a concept consists of a set of rules that describe objects. For example, a haiku is a concept, consisting of logic that can be used to classify poems. Meanwhile a fact like "Abraham Lincoln was born in 1809" is not a concept because it does not correspond to any set of generalizable rules.

What does it mean for a human to understand a concept? The full scope of possible ways to demonstrate understanding is enormous; for example, we may expect someone who understands the concept of a haiku to be able to define it, generate haikus about arbitrary topics, and categorize every poem as a haiku or otherwise. Asking people to enumerate every possible example of a concept is intractable. Instead, it typically suffices to demonstrate understanding with a few specific examples. For example, if someone can accurately define a haiku and provide a few examples, we would have confidence that they have understood the concept of a haiku.

Why is it reasonable to infer that people have understood a concept after only seeing a few examples? The key insight is that while there exist a theoretically very large number of ways in which humans might misunderstand a concept, only a limited number of these misunderstandings occur in practice. This is because people misunderstand concepts in structured ways. For example, if a person mistakenly believes that haikus follow a 5-8-5 syllabic structure, the examples of haikus they come up with will all be incorrect in the same way. This logic underlies why we use exams to test conceptual understanding in people: even though SAT and AP exams only consist of a tiny fraction of possible questions about a concept, the questions are structured so that conceptual understanding is necessary to achieve high scores. The space of human misunderstandings is predictable and sparse.

We formalize this notion by defining $\mathcal{X}$ as the set of all strings that are relevant to a concept: for example, a string can correspond to a possible definition of a concept or a possible example of one. Not every string that's relevant to a concept is a valid use of it. An **interpretation** of a concept is defined to be any function $f : \mathcal{X} \to \{0, 1\}$, where the output indicates whether the string is considered valid in the interpretation (0 for invalid, 1 for valid). There is a single correct interpretation, denoted by $f^*$. The set of possible ways for humans to interpret concepts is denoted by $\mathcal{F}_h$. Every function $f \in \mathcal{F}_h$ that is not equal to $f^*$ corresponds to a way in which a human might misunderstand a concept.

Consider one possible way $f \in \mathcal{F}_h$ that a human might interpret a concept. To evaluate whether a human has understood the concept correctly, we don't check whether $f(x) = f^*(x)$ across all strings $x \in X$; this is intractable. Instead, we use exam questions to isolate specific instances that serve as proofs of understanding. But what constitutes such a proof? The answer is revealed by this framework:

**1. Single-element keystones**

**2. Multi-element keystones**

*Figure 2.* A schematic representation of our conceptual framework. Rows represent different interpretations of a concept and columns represent different questions. Questions can either be **validly** or **invalidly** classified by an interpretation. The figure provides a stylized depiction of keystones, interpretations, and potemkins.

the sets of examples that are only interpreted correctly if a human's interpretation is correct.

Formally, we define a **keystone** set $\mathcal{S} \subseteq \mathcal{X}$ as a minimal set of instances such that if $f \in \mathcal{F}_h$ and $f(x) = f^*(x)$ for all $x \in \mathcal{S}$, then $f = f^*$. That is, if every example in the keystone set is interpreted correctly, it cannot be reconciled with any misconceived human notion of the concept. See Figure 2 for a visual depiction of a keystone set.

This approach shows why testing conceptual understanding in people is feasible. Testing understanding of a concept doesn't require testing people for all relevant examples. Instead, it only requires testing examples in a keystone set.

**Potemkins invalidate LLM benchmarks.** Humans are evaluated on tests that are designed to highlight keystones — questions that can only be answered correctly by a person who has fully understood a concept. Common benchmarks for evaluating LLMs include questions sourced from AP exams, SAT tests, medical examinations, and other standardized assessments (Hendrycks et al., 2020; Clark & Etzioni, 2016; Jin et al., 2021).

But how effective are tests that are designed for people when it comes to evaluating LLMs? To answer this question, define $\mathcal{F}_l$ as the set of ways for any possible LLM to interpret a concept, where each $f \in \mathcal{F}_l$ is an interpretation $f : \mathcal{X} \to \{0, 1\}$.[2]

**Definition 2.1.** An LLM has **potemkin understanding** if its interpretation satisfies $f(x) = f^*(x)$ for all $x$ in a keystone $S$, but $f(x) \neq f^*(x)$. In this case, we refer to any

---

[2]We don't impose a single way to translate an LLM's output into a statement about correctness; for example, an LLM might be instructed to return 0 if a particular instance is incorrect and 1 if it's correct.

$x$ such that $f(x) \neq f^*(x)$ as a **potemkin**.

In other words, potemkins arise when an LLM answers keystone questions correctly but does not have the correct interpretation of a concept. The prevalence of potemkins in LLMs is consequential for benchmarking:

**Result:** Keystones are a valid way to test LLMs if $\mathcal{F}_l = \mathcal{F}_h$.

**Corollary:** If an LLM displays a potemkin, it illustrates that keystones are invalid tests of LLM understanding.

These results imply that benchmarks based on keystone questions for people are invalid tests for LLMs if LLMs are capable of potemkin understanding. Suppose we were to test an LLM like we would a human: by testing whether it correctly answers the questions in a keystone set. For example, we may prompt it with the questions on an AP exam and measure whether it answers them correctly. A human that performs well on these questions must have the correct interpretation $f^*$ because their interpretations are limited to $\mathcal{F}_h$. However, an LLM that has not understood the concept may still perform well on the exam if $\mathcal{F}_l \neq F_l$; that is, if it is capable of misinterpreting a concept in ways that do not mirror human misinterpretations.

We note that under the hold-out principle, performance on benchmark questions still guarantees performance on other questions drawn from the same distribution; if an LLM scores 95% accuracy on an i.i.d. sample of AP questions, it is expected to score 95% on a separate sample. However, analogous to how the hold-out principle only holds under i.i.d. data assumptions, success on keystone questions only guarantees success on other relevant tasks under a "no potemkins" assumption. It is on these other tasks — not standardized tests — that we typically care about LLM performance in the real world.

Thus, to evaluate whether LLM benchmarks work, we must first determine the prevalence of potemkins. The remainder of the paper is dedicated to developing procedures to quantify the prevalence of potemkins.

## 3. A Benchmark Dataset for Potemkins

This paper presents two procedures for measuring the prevalence of potemkins in large language models (LLMs). This section describes one procedure, based on a benchmark dataset we collect that measures a specific kind of potemkin failure: a divide between describing and applying concepts. Specifically, we construct a dataset spanning 32 concepts from 3 distinct domains: literary techniques, game theory, and psychological biases, collecting 3, 159 labeled data points. We find that even when models can correctly define a concept, they often fail to accurately apply it in classification, generation, and editing tasks. All collected data, annotations, and analysis are made publicly available at the Potemkin Benchmark Repository.[3]

### 3.1. Benchmark motivation

We design a benchmark that measures a specific kind of potemkin. Observe that any human who can answer a keystone about a concept must be able to correctly *use* that concept in a concrete instance. This is because, by definition, keystone success in humans indicates correct concept understanding. Thus, we identify potemkins in LLMs as instances where (1) the LLM can correctly answer a keystone about a concept but (2) it fails to correctly use that same concept in a concrete instance.

What might we use as our keystone? A common keystone for a concept is **definitions**; we have faith that humans who can clearly define the concept of a haiku have understood haikus. Thus, a potemkin occurs when an LLM that can define a concept correctly cannot use it.

What would it mean for an LLM to be able to use a concept in a concrete instance? We consider three such tasks, each that offers a unique perspective for measuring potemkins. One task is **classification**: answering whether an example is a correct application of a concept. Another task is **generation**: producing an instance of a concept that adheres to specific constraints. The last task we consider is **editing**: modifying an example so that it either belongs or doesn't belong to a concept. We provide more details for these tasks in Section 3.2; see Appendix Section A for a visual representation of our experiment.

What it means to use a concept depends on the domain being tested. We choose concepts from a diverse array of domains:

---

[3] https://github.com/MarinaMancoridis/
PotemkinBenchmark.git

**literary techniques**, **game theory**, and **psychological biases**. These domains together span generative linguistics, formal constructs, and human understanding. We examine concepts like "*analogy*" in literary techniques, "*Pareto optimality*" in game theory, and "*sunk cost fallacy*" in psychological biases. We explore a total of 32 distinct concepts within the domains, with a full list provided in Appendix Section B. Given the diversity of our concepts and tasks, evidence of potemkins in our analysis would suggest not an isolated issue but a systemic category of failure in LLMs.

### 3.2. Benchmark construction.

We construct datasets to evaluate concept explanation and concept use in each of the three domains. To achieve this, we construct datasets for each of the three domains, varying generation and evaluation methods to enhance the robustness of our findings. Collecting data or validating responses from some domains requires hand-labeling, while others are automatic. To ensure high-quality annotations, we rely on a mix of domain experts and paper authors to evaluate model responses. Our analysis spans the following 7 models: Llama-3.3 (70B), GPT-4o, Gemini-2.0 (Flash), Claude-3.5 (Sonnet), DeepSeek-V3, DeepSeek-R1, and Qwen2-VL (72B). Model names are abbreviated in subsequent tables. Below, we describe our data collection process for each of the four tasks in our evaluation framework.

**Definition.** To assess whether LLMs can explain concepts, we prompted models to define a concept in a given domain. We prompted each of 7 models to define each of 32 concepts, resulting in a total of 224 generated definitions across domains. We evaluated the definitions ourselves, as some of the concepts required specialized knowledge to evaluate accurately. For example, evaluating the accuracy of a definition for the literary techniques concept of a "Shakespearean sonnet" required confirming that it accurately described the ABAB CDCD EFEF GG rhyming scheme and specified that the poem must be entirely written in iambic pentameter.

**Classification.** Models must determine if presented examples are valid instances of a given concept. The model is given an instance and asked: "Is the following example a true instance of the concept $c$?" For example, to assess the model's grasp of the concept of "slant rhyme", we could present pairs of words—some that rhyme and others that do not—and ask the model to evaluate whether they qualify.

Evaluating a model's ability to classify concepts requires creating positive and negative examples. We used distinct data generation approaches per domain. For *literary techniques*, we crafted original examples and collected online examples from recent poetry competitions (ensuring they post-date each LLM's training cutoff). As the *game theory* domain requires specialized knowledge, we recruited Economics PhD students to produce true and false instances.

For the *psychological biases* domain, we gathered 40 text responses from Reddit's *"r/AmIOverreacting"* thread, annotated by expert behavioral scientists. Models classified each example, and we compared their outputs against our labeled ground truth. Overall, we generated 2,030 annotations. For further details, see Appendix Sections C and D.

**Constrained generation.** This task assesses a model's ability to use concepts by requiring it to generate examples adhering to specific constraints. This tests the model's capacity to flexibly apply concepts within structured boundaries. For instance, for the concept of "strict dominance" in game theory, we might ask the model to construct an example specifying constraints like the number of players or which players have strictly dominant strategies.

Performance measurement involved three steps: (1) defining constraints for each concept, (2) generating model responses, and (3) annotating responses to assess constraint adherence. Constraints were defined by the paper authors. Labels varied by domain: literature labels came from the authors, game theory labels were generated automatically, and psychological biases labels were determined by majority expert consensus. Overall, 224 model responses were evaluated. See Appendix Sections E and F for detailed specifications.

**Editing.** This task evaluates a model's ability to use concepts by requiring it to identify modifications that could transform an instance into either a true or false example of a given concept. For instance, we might present the model with a partially obscured haiku—where a section of the poem is missing—and ask what could be added to complete the poem and ensure it qualifies as a true instance of a haiku. To assess performance, we prompt the model with an instance to edit and evaluate its response. Our prompting and evaluation strategies were tailored to each domain. In total, we gathered 791 annotations of model responses. For further specifications, see in Appendix Section G.

### 3.3. Results

We analyze 7 large language models across 32 concepts. These models were chosen for their popularity and range of developers and sizes. We collect inferences using APIs from OpenAI, Together.AI, Anthropic, and Google.

For each (model, concept) pair, we first determine whether the model provides a correct definition. If so, we evaluate its accuracy on the three additional tasks: classification, generation, and editing. Responses are labeled as correct or incorrect according to our framework specifications.

We measure the **potemkin rate** exhibited by models. We define the potemkin rate of a model as the proportion of questions that the model solves incorrectly when it solves a keystone correctly. Formally, we compute this as $1-$ task accuracy conditional on keystone success. For tasks with a

| Model | Potemkin rate, as measured by: | | |
| | Classify | Generate | Edit |
| --- | --- | --- | --- |
| Llama-3.3 | 0.57 (0.06) | 0.43 (0.09) | 0.36 (0.05) |
| Claude-3.5 | 0.49 (0.05) | 0.23 (0.08) | 0.29 (0.04) |
| GPT-4o | 0.53 (0.05) | 0.38 (0.09) | 0.35 (0.05) |
| Gemini-2.0 | 0.54 (0.05) | 0.41 (0.09) | 0.43 (0.05) |
| DeepSeek-V3 | 0.57 (0.05) | 0.38 (0.09) | 0.36 (0.05) |
| DeepSeek-R1 | 0.47 (0.05) | 0.39 (0.09) | 0.52 (0.05) |
| Qwen2-VL | 0.66 (0.06) | 0.62 (0.09) | 0.52 (0.05) |
| **Overall** | **0.44 (0.02)** | **0.40 (0.03)** | **0.40 (0.02)** |

Table 1. *Potemkin rate on classify, generate, and edit tasks*, conditioned on correctly defining each concept. We scale the classification values so that chance-level performance corrsponds with a potemkin rate of 1. Standard errors are in parentheses.

chance accuracy of $0.50$, we scale this value by a factor of $2$, so that a potemkin rate of $1$ corresponds to chance-level performance.

Our findings reveal high potemkin rates across all models and domains, as summarized in Table 1. Models define concepts correctly 94.2% of the time. However, conditioned on correct definitions, their performance sharply decreases when tasked with *using* those concepts, as exhibited by the high potemkin rates in the table.

While performance varies slightly across models and tasks, we find that ***potemkins are ubiquitous*** across all models, concepts, and domains that we analyzed. See Table 6 (Appendix Section I) for further details of our results. Further examples of potemkins are provided in Figure 3.

**Discussion.** We raise two points of discussion related to our benchmark. One possible concern is that while our benchmark only uses single definition questions as part of the keystone set, in reality keystones may include multiple questions, not only about defining the concept. For example, we may not have faith that a student who defines the quadratic formula truly understands what it is until we see them apply it a few times.

To address this, we conduct a supplementary analysis that simulates performance when keystones contain additional questions. Specifically, for each concept, we consider keystone sets that require not only the correct definition but also $k$ correct responses to classification ("use") questions. We perform a simulation exercises that measures, among concepts for which an LLM answers the expanded set of keystones correctly, how well it performs on other "use" questions. Specifically, we say a model has "understood a concept" if it answers 10 additional "use" questions after answering keystones correctly. Figure 4 shows how concept understanding varies with the size of the keystone set (only

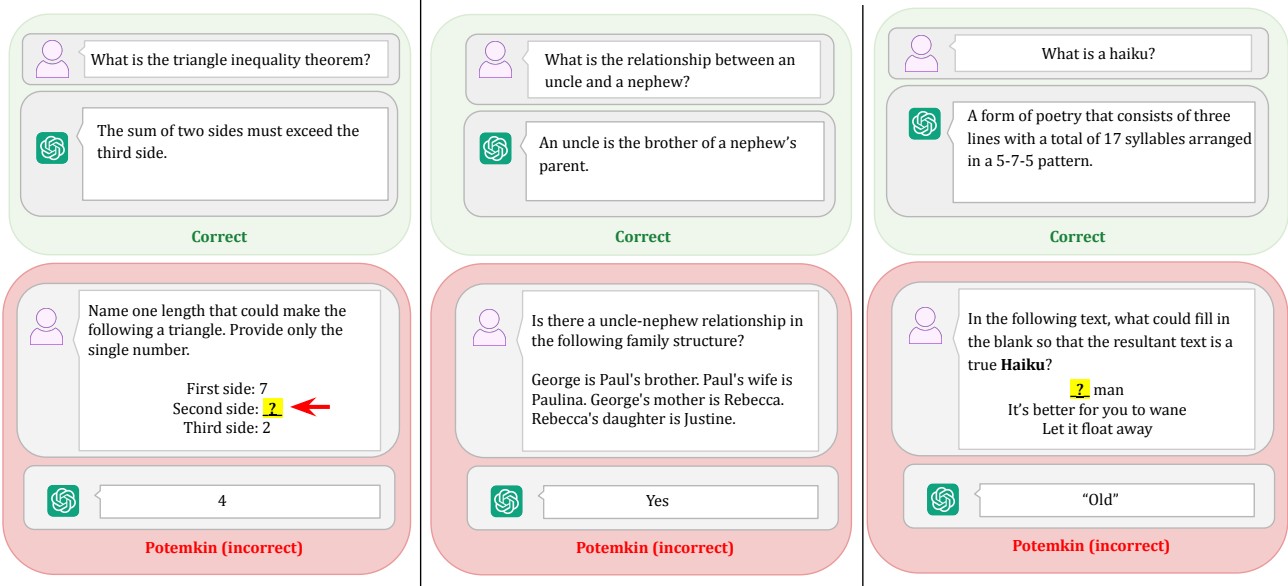

*Figure 3. Examples of potemkins.* In each example, GPT-4o correctly explains a concept but fails to correctly use it.

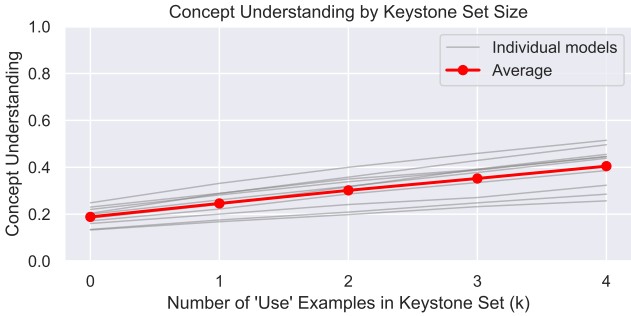

*Figure 4. Impact of expanding keystone sets on concept understanding values.* We extend the keystone from a single definition to include multiple classification ("use") examples. Conditioning on a correct definition, we incrementally require additional correct classification responses. Concept understanding values increase modestly, from $0.19$ when $k = 0$ to $0.40$ when $k = 4$.

including models that outperform chance). Expanding the keystone set yields only modest performance gains.

Another concern is whether the *use* tasks we consider in the benchmark are too hard such that humans would fail these as well. Qualitatively, we find this is not the case; Appendix J shows examples of failures that humans who understand the concept would not make.

## 4. Automatically Evaluating Potemkins

In this section we present a different, automated procedure for evaluating the presence of potemkins.

### 4.1. Warmup: Incoherence

Section 3 demonstrates that potemkin understanding is ubiquitous in LLMs. There are two possibilities for why this might be the case. One possibility is that the LLMs have slightly misaligned but internally consistent understanding of concepts. Another possibility is that their conceptual grasp is incoherent, with conflicting notions of the same idea.

To distinguish between these two cases, we test specifically for conceptual incoherence within models. We measure incoherence in two steps. First, we prompt a model to generate either an instance or a non-instance of a specific concept (e.g., producing an example of a slant rhyme). Then, we present the model's generated output back to it (in a separate query), asking whether this output is indeed an instance of the concept. In the slant rhyme example, this means testing if the model recognizes its own example as a slant rhyme. Figure 5 summarizes this procedure.

We quantify incoherence by first calculating the percentage of cases where the model's initial generation does not match its subsequent classification. Since random-chance accuracy is $0.50$, we then multiply this value 2, rescaling scores such that 0 indicates no incoherence and 1 indicates as-good-as-random performance. See Appendix Section K for more details on our data collection process.

The results of our analysis are presented in the first column of Table 2. We observe incoherence across all examined models, concepts, and domains, with scores ranging from $0.02$ to $0.64$. Although these scores are better than random, they nonetheless indicate substantial limitations in models'

**Step 1**

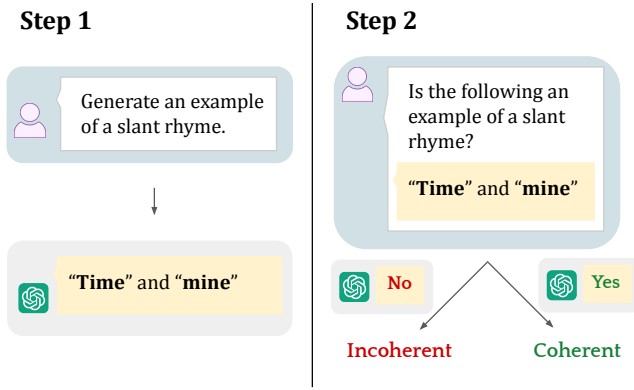

**Step 2**

*Figure 5. Illustration of the two-step method for evaluating incoherence in models.* In the first step, the model generates an instance or non-instance of a given concept. In the second step, the model evaluates whether the instance it generated is a true or false example of the concept.

| Model | Incoherence | Potemkin rate *(lower bound)* |
|---|---|---|
| Llama-3.3 | 0.18 (0.04) | 0.82 (0.02) |
| Claude-3.5 | 0.62 (0.06) | 0.36 (0.02) |
| GPT-4o | 0.64 (0.06) | 0.46 (0.06) |
| GPT-o1-mini | 0.16 (0.03) | 0.66 (0.02) |
| GPT-o3-mini | 0.02 (0.01) | 0.66 (0.04) |
| Gemini-2.0 | 0.10 (0.02) | 0.86 (0.02) |
| DeepSeek-V3 | 0.12 (0.02) | 0.38 (0.02) |
| DeepSeek-R1 | 0.04 (0.02) | 0.50 (0.02) |
| Qwen2-VL | 0.12 (0.02) | 0.82 (0.00) |
| **Overall** | **0.22 (0.01)** | **0.62 (0.01)** |

*Table 2. Incoherence scores and potemkin rates across models.* An incoherence score of $0$ indicates perfect performance and a score of $1$ indicates good-as-random performance. Potemkin rate is defined as $1-$ accuracy, multiplied by $2$ (since random-chance accuracy on this task is $0.5$, implying a baseline potemkin rate of $0.5$). The automatic evaluation procedure provides a lower bound on potemkin rate. Standard errors are in parentheses.

ability to consistently evaluate their own outputs. This indicates that conceptual misunderstandings arise not only from misconceiving concepts, but also from inconsistently using them. A detailed breakdown of incoherence scores by domain is provided in Appendix Section L.

### 4.2. A lower bound on potemkin rates.

Our incoherence analysis used LLM self-grading as a way to measure concept misunderstanding. This insight motivates another, more general automated procedure for measuring potemkins. The idea is as follows: if an LLM understands a concept, its answers to new questions about that concept should be judged as being correct *by the same LLM*. How often the LLM judge disagrees with its original answers reveals the prevalence of potemkins.

Specifically, we prompt an LLM with questions from a benchmark and automatically grade whether it is correct. If it answers correctly, we prompt it to generate other questions that rely on the same concept. Then, for each question, we prompt the LLM to answer the question correctly and then re-prompt the same LLM to grade its response. Whenever the judge's response deviates from the expected response, it indicates an example of a potemkin. See Figure 6 for an example of this procedure.

Importantly, this approach provides a *lower-bound* on the rate of potemkins. To see why, consider the scenarios in which the LLM judge grades a response as incorrect. If the LLM judge correctly grades an incorrect response as incorrect, that LLM exhibited misunderstanding in responding to the original question. If the LLM judge mistakenly grades a correct response as incorrect, it still reveals conceptual misunderstanding in the judge: a judge that understood the concept would've graded it correctly. The reason it's a lower

bound is that the judge may grade an incorrect response as correct, which would be an example of conceptual misunderstanding but isn't captured by this metric. Similarly, generating questions that are unrelated to a concept would also demonstrate misunderstanding, although qualitatively we do not find this to be the case.

We perform this analysis for the same models as above using questions sampled from MMLU (Hendrycks et al., 2020). We prompt LLMs to generate 5 related questions for each conceptual question it answers correctly and then prompt it to respond to each question. In order to avoid sycophancy in the judge (Malmqvist, 2024) (e.g. an LLM may always grade responses as correct), we also prompt the LLM to generate a slightly incorrect response and also re-prompt the LLM to grade it (see Appendix Section M for examples of prompts). Results are reported in Table 2. Our reported potemkin rate is scaled so that 1 corresponds to chance accuracy and 0 corresponds to perfect performance. Despite the procedure only providing a lower bound, the potemkin rate is high: 0.62. These results corroborate the benchmark findings from Section 3.

## 5. Related Work

**Providing a method to validate benchmarks.** Researchers caution that high scores on standard NLP benchmarks may not be informative of true understanding, highlighting that the "leaderboard mentality" overestimates perceptions of a model's capabilities (Bowman & Dahl, 2021; Mitchell, 2021; Church & Hestness, 2019; Ribeiro et al., 2020; Recht et al., 2019; Raji et al., 2021). Benchmarking raises the

**1.  LLM answers question correctly**

If two people share the same great-grandparents but different grandparents, they are:

A) First cousins
B) Second cousins
C) First cousins once removed
D) Third cousins

Claude Sonnet 4

B) Second cousins

Correct

**2.  Ask LLM to generate related question and answer it correctly**

Generate a related question and answer it.

**Question**: At a family reunion, you meet someone who says "My dad's grandpa and your mom's grandpa were brothers." What's your relationship?

**Answer**: You would be **third cousins**.

**3.  Ask the same LLM to grade its answer**

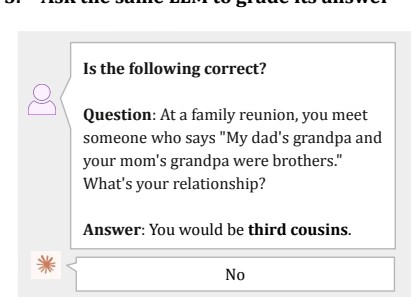

**Is the following correct?**

**Question**: At a family reunion, you meet someone who says "My dad's grandpa and your mom's grandpa were brothers." What's your relationship?

**Answer**: You would be **third cousins**.

No

Incoherent

*Figure 6. Example of a potemkin documented by our automatic evaluation procedure.* In this example, the language model fails to agree with its own answer for a question that it generated about the topic of second cousins.

challenge of *construct validity*—the degree to which a test measures the concept it claims to measure.

Many challenges undermine the reliability of benchmarks as measures of conceptual understanding. Benchmarks are frequently inadequately developed due to undervalued data collection efforts (Gebru et al., 2021; Paullada et al., 2021; Sambasivan et al., 2021). High scores may instead reflect superficial shortcuts (Geirhos et al., 2020; Lapuschkin et al., 2019), while benchmarks often lack sufficient statistical power (Card et al., 2020) and aggregate metrics obscure critical failures (Wu et al., 2019). These challenges may reflect broader fallacies in measuring intelligence, such as assuming intelligence can be neatly placed on a linear scale or task difficulty uniformly gauged (Mitchell, 2021).

Efforts to improve evaluation approaches include tests of robustness to noise (Belinkov & Bisk, 2017; Rychalska et al., 2019), adversarial perturbations (Ribeiro et al., 2018; Iyyer et al., 2018), logical consistency checks (Ribeiro et al., 2019), focusing on explanations (Ribeiro et al., 2016), and behavioral evaluations through frameworks such as Check-List (Ribeiro et al., 2020). There has been a shift from static benchmarks to dynamic, real-world interaction-based assessments, exemplified by ChatBench (Chang et al., 2025), WildChat (Zhao et al., 2024), ChatbotArena (Chiang et al., 2024), and the human-in-the-loop adversarial framework Dynabench (Kiela et al., 2021). Additionally, best practices in benchmarking have been explored through initiatives like BetterBench (Reuel et al., 2024) and comprehensive, multi-metric evaluation methods (Liang et al., 2022).

Our paper introduces a framework that clarifies when benchmarks validly measure true concept understanding in LLMs: when there are no potemkins. We provide two procedures for finding potemkins and show potemkins to be ubiquitous in our analysis. The high potemkin rates that we observe invalidate using existing benchmarks as measures of understanding.

**Providing a framework for evaluating depth of concept understanding.** Researchers have documented numerous *failure modes* in large language models, including limitations in linguistic reasoning (Dagan et al., 2010; Bowman et al., 2015; Dentella et al., 2024; Arkoudas, 2023; Dickinson & Meurers, 2003), common-sense problem solving (Shwartz & Choi, 2020), planning (Valmeekam et al., 2023), alignment with human generalization (Vafa et al., 2024b), factual recall and knowledge representation (Meng et al., 2022; Geva et al., 2020; 2022; 2023; Tam et al., 2022; Petroni et al., 2019; Zhu et al., 2020; Mitchell et al., 2021; Yao et al., 2022), distractability (Shi et al., 2023), and coherent world-model formation (Ivanova et al., 2024; Ha & Schmidhuber, 2018; Vafa et al., 2024a). They have identified *logical inconsistencies* through methods including reversed claims (Saba, 2024; Berglund et al., 2023), decision-making inconsistencies (Fluri et al., 2024), factual correctness in paraphrases (Elazar et al., 2021), compositional tasks (Press et al., 2022), negation handling (Hosseini et al., 2021), common human misconceptions (Lin et al., 2021), semantic equivalence (Jang et al., 2021), consistency benchmarks (Jang et al., 2022; Jang & Lukasiewicz, 2023), and validation-generation alignment (Li et al., 2023).

Our work highlights that models may comprehend concepts in fundamentally different ways than humans. Potemkins reveal modes of conceptual understanding that humans, by construction, cannot exhibit. Our work frames prior methods of evaluating concept understanding as implicit tests for potemkins, and proposes an explicit, systematic approach to uncovering these discrepancies. Potemkins are to conceptual knowledge what hallucinations are to factual knowledge—hallucinations fabricate false facts; potemkins fabricate false conceptual coherence. Yet potemkins pose a greater challenge: hallucinations can be exposed through fact-checking, but potemkins require unraveling subtle inconsistencies in a model's apparent understanding. By identifying potemkins, we introduce implementable frameworks

to assess benchmark validity and conceptual depth, transforming the abstract notion of understanding into actionable tools that can be leveraged for future model improvement.

## 6. Conclusion

In this paper, we introduce the phenomenon of *potemkin understanding*— a failure mode of large language models (LLMs) whereby apparent comprehension revealed by successful benchmark performance is undermined by non-human patterns of misunderstanding. We began by formalizing a framework for evaluating when benchmarks designed for humans serve as valid tests for understanding in LLMs. This framework revealed a crucial dependency: benchmarks are only valid if LLMs misunderstand concepts in the same structured ways humans do. If models deviate from human-like misunderstandings, they can still correctly answer keystone questions without truly understanding the underlying concept, producing potemkins.

Through two complementary empirical procedures—one leveraging a novel benchmark dataset across literary techniques, game theory, and psychological biases, and the other employing an automated evaluation strategy—we quantified the prevalence of potemkins across a variety of tasks, concepts, domains, and models. Both procedures reveal high potemkin rates, even in models that appear highly capable by conventional benchmark standards. Our tests for incoherence reveal that models contain conflicting representations of the same idea.

Our approach has limitations that suggest avenues for further exploration. The benchmark datasets we developed, while extensive, are not exhaustive. Additional datasets spanning a broader range of concepts, domains, and abstraction levels could enable a more comprehensive identification of potemkins. There is also significant potential in methodologies explicitly designed to detect and reduce potemkin rates in language models. Systematically integrating techniques for potemkin detection and mitigation into existing model training and evaluation pipelines represents an especially promising direction for future research.

## Acknowledgments

Marina Mancoridis is supported by the MIT Presidential Graduate Fellowship. Keyon Vafa is supported by the Harvard Data Science Initiative. We also thank Steven Ma, Cassidy Shubatt, and Charlotte Siegmann for helpful comments and feedback.

## Impact Statement

Our work puts forth formalizations and benchmarks to assess concept understanding in large language models (LLMs). We document *potemkin understanding* as an overlooked but pervasive category of failure in LLMs. By demonstrating that models can correctly answer keystone questions without true concept comprehension, our findings compel a re-examination of the validity of benchmarks widely trusted in AI development. The framework and empirical tools we provide allow researchers and practitioners to systematically identify and address potemkins, paving the way toward the development of deeper and more reliable concept understanding.

This paper presents a dataset using data collected from human surveys. We received an IRB review and exemption for this study.

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

# A. Illustration of the Benchmark Design

Figure 7 provides a schematic representation of our experimental framework, with examples from each of our three domains. The yellow examples correspond to the concept of a haiku, in the literary techniques domain. The green examples correspond to the concept of a mixed strategy Nash Equilibrium, in the game theory domain. Finally, the gray examples correspond to the concept of a sunk cost fallacy, in the psychological biases domain.

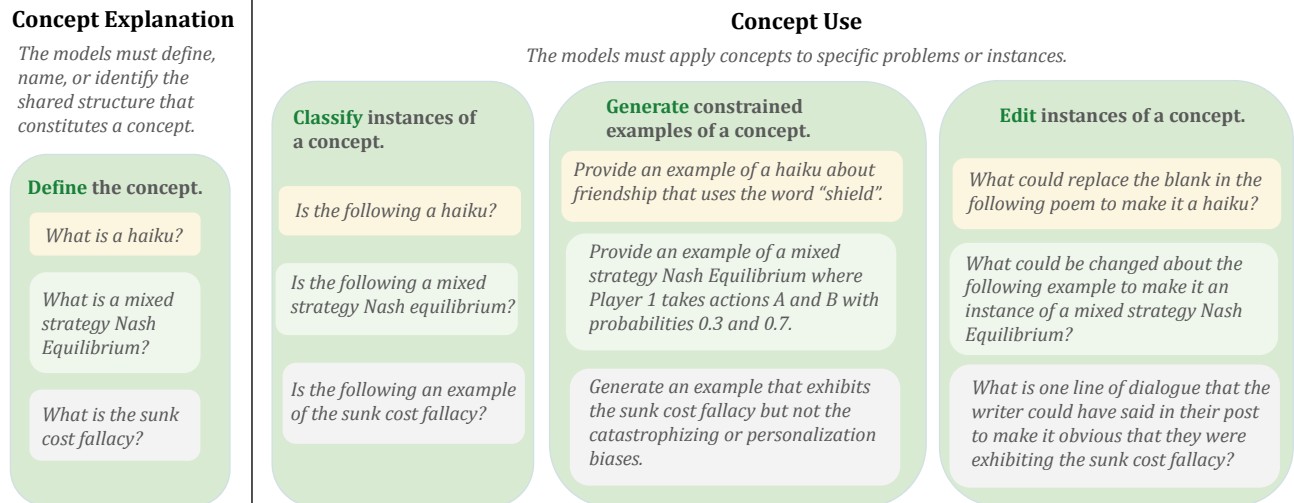

Figure 7. *Our framework for evaluating concept understanding, including examples.* We evaluate concept explanation by asking the model to define the concept. We measure concept use by giving the model classification, constrained generation, and edit tasks.

# B. Concept Choices

Within each domain, we selected a diverse set of concepts for our main analysis, totaling 32 concepts: 12 from literary techniques, 9 from game theory, and 11 from psychological biases. **Table 3** lists all concepts with abbreviated definitions.

| Domain | Concept | Definition |
|---|---|---|
| **Literary Techniques** | Haiku | A 3-line poem with a 5-7-5 syllable structure |
| | Shakespearean Sonnet | A 14-line poem with ABABCDCDEFEFGG rhyme scheme, written in iambic pentameter |
| | Analogy | Comparison highlighting similarities between two things |
| | Paradox | A seemingly contradictory but true statement |
| | Anacoluthon | Sudden break in grammatical structure for effect |
| | Asyndeton | Omission of conjunctions for a concise effect |
| | Hyperbaton | Reordering words for emphasis |
| | Synesis | Agreement of meaning over grammar |
| | Accismus | Feigned refusal of something desired |
| | Slant Rhyme | Near rhyme with similar but not identical sounds |
| | Enthymeme | An argument with an implied premise |
| | Anapest | Metrical foot with two unstressed and one stressed syllable |
| **Game Theory** | Strict Dominance | A strategy always better than others, regardless of opponents |
| | Iterated Dominance | Successive elimination of dominated strategies |
| | Weak Dominance | A strategy at least as good as others and better in some cases |
| | Pure Nash Equilibrium | Players choose specific strategies with no incentive to deviate |
| | Mixed Strategy Nash Equilibrium | Players randomize strategies with no incentive to deviate |
| | Pareto Optimality | No player can be better off without making another worse off |
| | Best Response | Choosing your payoff-maximizing response to others' actions |
| | Zero-Sum Game | One player's gain equals another's loss |
| | Symmetric Game | Payoffs depend on strategies, not players |
| **Psychological Biases** | Fundamental Attribution Error | Explaining others' behavior by personality, not situation |
| | Black and White Thinking | Viewing situations in extreme, all-or-nothing terms |
| | Sunk Cost Fallacy | Irrationally continuing an endeavor due to past investments |
| | IKEA Effect | Valuing something more because you assembled it yourself |
| | Pseudocertainty Effect | Treating conditional outcomes as certain |
| | Endowment Effect | Valuing something more just because you own it |
| | Naive Cynicism | Assuming others' motives are more self-interested than your own |
| | Normalcy Bias | Underestimating risk by assuming things will continue as usual |
| | Spotlight Effect | Overestimating how much others notice you |
| | Illusory Superiority | Overestimating your abilities compared to others |
| | Catastrophizing | Assuming the worst possible outcome will occur |

*Table 3.* Concepts and their definitions grouped by domain.

# C. Specifications of the Classification Task

After assembling the dataset of true and false instances of each concept, we automated the evaluation of model classifications by directly comparing the model's outputs to our labeled ground truth. Across all of our models and domains, we collected a total of **2,030 annotations**. These annotations were collected as follows:

1. **Literary techniques.** We evaluate 7 models across 12 concepts. For each concept, we consider 10 examples—5 true instances and 5 false instances. This results in a total of **840 annotations**.

2. **Game theory.** We evaluate 7 models across 9 concepts. For each concept, we consider 10 examples—5 true instances and 5 false instances. This results in a total of **630 annotations**.

3. **Psychological biases.** We gathered 40 examples from posts on the subreddit `r/AmIOverreacting`, each annotated for the presence of 2 (from a list of 11) psychological biases. Labels were determined by majority vote among trained psychologists recruited via Upwork. Each psychologist evaluated 10 posts, providing labels for 2 biases per post. For more details on the survey design, see Appendix Section D. Every (post, bias) pair was labeled by 3 psychologists, and we retained the majority label. Subsequently, we asked each of 7 models to classify each post for both biases. This results in a total of **560 annotations**.

For our analysis, we exclude responses missing the specified format (e.g., missing strings like "ANSWER: _").

## D. Classification Task: Survey for Expert Annotation of Psychological Biases

To establish reliable ground-truth labels for psychological biases, we conducted an expert annotation survey with psychologists recruited via Upwork. Experts were shown posts from the subreddit r/AmIOverreacting and asked to determine whether each of two psychological biases was present in each post. Full specifications of the annotation collection process can be found in Appendix Section C. Figure 8 shows an example screen from the expert annotation survey.

**Post 4/10**

Read the following post from Reddit.

My(19M) girlfriend(19F) is a very private person. She never posts anything on social media, and usually she tends to be very secretive, like I found that she kind of drives conversations towards talking about me, and my interests. But it's really hard to learn more about her because she doesn't let me. She's like this everyone except for her best friend Marie. Above all else, she is super super possessive about her phone, she's never showed me any of her photos or anything, she always takes her phone with her, she has one of those advanced passcodes, and she always turns her phone away from my direction, and just acts a little sus in general. This has caused me lots of mental torture, as I don't know what's going on, if she even likes me, I keep on thinking I'm not good enough for her, and it just feels weird that she's so cautious around me.

**Bias 1:**

Consider the following bias: Naive Cynicism.

Here is the definition: A cognitive bias where individuals assume that others' actions or beliefs are driven primarily by selfish motives, biases, or hidden agendas, while considering themselves relatively objective and unbiased.

Does the author of the post exhibit the bias defined above?

○ Yes

○ No

**Bias 2:**

Consider the following bias: Spotlight Effect.

Here is the definition: A cognitive bias where individuals overestimate how much others notice or pay attention to their appearance, behavior, or actions, believing they are under a 'spotlight.'

Does the author of the post exhibit the bias defined above?

○ Yes

○ No

*Figure 8.* Example screen from the expert annotation survey used to label psychological biases.

# E. Specifications of the Constrained Generation Task

## E.1. Literary Techniques

In the domain of *literary techniques*, we required model outputs to adhere to three types of constraints. The first constraint is *semantic*: the example must involve a theme such as "gratitude" or "friendship." The second constraint is *form*: the example must comply with a linguistic constraint like using dialogue or starting with a monosyllabic word. The last constraint is *lexical*: the example must incorporate a specific word, such as "gravity" or "wake." Each prompt was constructed by randomly selecting one constraint from each category. For example, one prompt might be: "*Write an example involving friendship. The example must use dialogue and include the word 'gravity'*".

To generate these constraints, we prompted GPT-4o to produce a list of 20 constraints for each of the three categories, totaling 60 unique constraints. The full list of constraints is shown in Table 4. We consider 7 models and 12 concepts. In our analysis, the paper authors annotate one responses for each (concept, model) pair, resulting in **84 annotations**.

| Semantic | Form | Lexical |
|---|---|---|
| Nature | Start with a monosyllabic word. | Father |
| Childhood memories | Start with a word that is not monosyllabic. | Red |
| A significant event | Start with a word that begins with the letter "M". | Order |
| Love | Start with a word that begins with the letter "I". | Press |
| Happiness | Use alliteration at least once. | Jump |
| Friendship | End with a one-vowel word. | Mother |
| Overcoming adversity | Use dialogue. | Run |
| Gratitude | Do not use dialogue. | Gravity |
| Change | Do not use any adjectives. | Shine |
| Hope | Do not use any adverbs. | Wonder |
| Moments in time | Make every sentence start with a verb. | Just |
| A meaningful object | Use the first-person perspective. | Justice |
| A meaningful person | Use the second-person perspective. | King |
| The earth | Use the third-person perspective. | Earth |
| A hobby | Include a sentence that contains a question. | Sunshine |
| Your dreams | Include a repeated phrase. | Light |
| Loss | Have at least two sentences or lines that start with the same word. | Wake |
| Your fears | Use a comma in the first sentence or line. | Friend |
| Feeling lost | Do not use any pronouns. | Nectar |
| Anger | Make the result in the past tense. | Love |

*Table 4. A full list of constraints used for the literary techniques domain in the constrained generation task.* We generated model prompts by **independently sampling one random constraint from each column**.

## E.2. Game Theory

In the domain of *game theory*, the constraints differed depending on the concept being tested. For instance, for the concept of "strict dominance," a constraint could specify whether one or both players must have a strictly dominant strategy in the example. Similarly, for the concept of "Pareto optimality," a constraint might require that all numbers in the payoff matrix be unique. We considered 7 models and 9 concepts. We generated one response for each (concept, model) pair, resulting in **63 annotations**. All annotations were generated by custom, automated evaluators.

| Concept Group | Constraint Prompt |
| --- | --- |
| Strict Dominance | There is a $x$ strategy for (one/both) players. |
| Weak Dominance | In the (row/column) that is $x$ for Player (1/2) receives a payoff of $a$ if the other player takes the corresponding actions. |
| Pure Strategy Nash Equilibrium | In the payoff matrix, all numbers must be unique. |
| Iterated Dominance | Start with an $a \times b$ matrix and end with a $c \times d$ matrix. |
| Zero-Sum Game | If Player (1/2) takes the $f$ action, they should receive a payoff of $(a, b, c)$ if the other player takes the corresponding actions. In the payoff matrix, all numbers must be unique. |
| Mixed Strategy Nash Equilibrium | In the mixed strategy Nash equilibrium, Player (1/2) should take Action 1 with a probability of $p$ and Action 2 with a probability of $1 - p$. |
| Pareto Optimality | You must generate a 3x3 matrix where there are exactly $e$ distinct Pareto optimal solutions. In the payoff matrix, all numbers must be unique. |
| Best Response | In the game, imagine Player 1 chooses the first action with a probability of $p_1$, the second action with a probability of $p_2$, and the third action with a probability of $p_3$. Player 2's best response to this action should be the $f$ action. |
| Symmetric Game | In the payoff matrix (combined for both players), all numbers must be unique, regardless of which Player the payoff is associated with. |

*Table 5. A full list of constraints used for the game theory domain in the constrained generation task.* We generated model prompts differently for each concept group, applying all respective constraint prompts. When generating prompts, we randomly chose between items in parentheses, replaced items $x$ with versions of the concept name, probabilities $p$ with single-point decimal values between 0 and 1, values $a$ through $d$ with integers $\in [1, 9]$, value $e$ with an integer $\in [1, 5]$, and action $f \in \{1\text{st}, 2\text{nd}, 3\text{rd}\}$. Note that if applicable, the uniqueness constraints are present in each generation prompt with a probability of $\frac{1}{2}$.

## E.3. Psychological Biases

In the *psychological biases* domain, models were tasked with generating examples illustrating two specified biases while explicitly avoiding a third, randomly selected bias. Six expert psychologists, recruited via Upwork, annotated the resulting model responses using a Qualtrics survey. Each expert evaluated responses from all 7 models across 2 concepts per model. For more details on the Qualtrics survey, see Appendix Section F. We obtained one response for each combination of 11 concepts and 7 models, resulting in **77 annotations**.

## F. Generation Task: Survey for Expert Annotation of Psychological Biases

To establish reliable ground-truth labels for psychological biases, we conducted an expert annotation survey with professional psychologists recruited via Upwork. Experts were shown inferences where models were tasked with generating examples illustrating two specified biases while explicitly avoiding a third, randomly selected bias. Full specifications of the annotation collection process can be found in Appendix Section E. Figure 9 shows an example screen from the expert annotation survey.

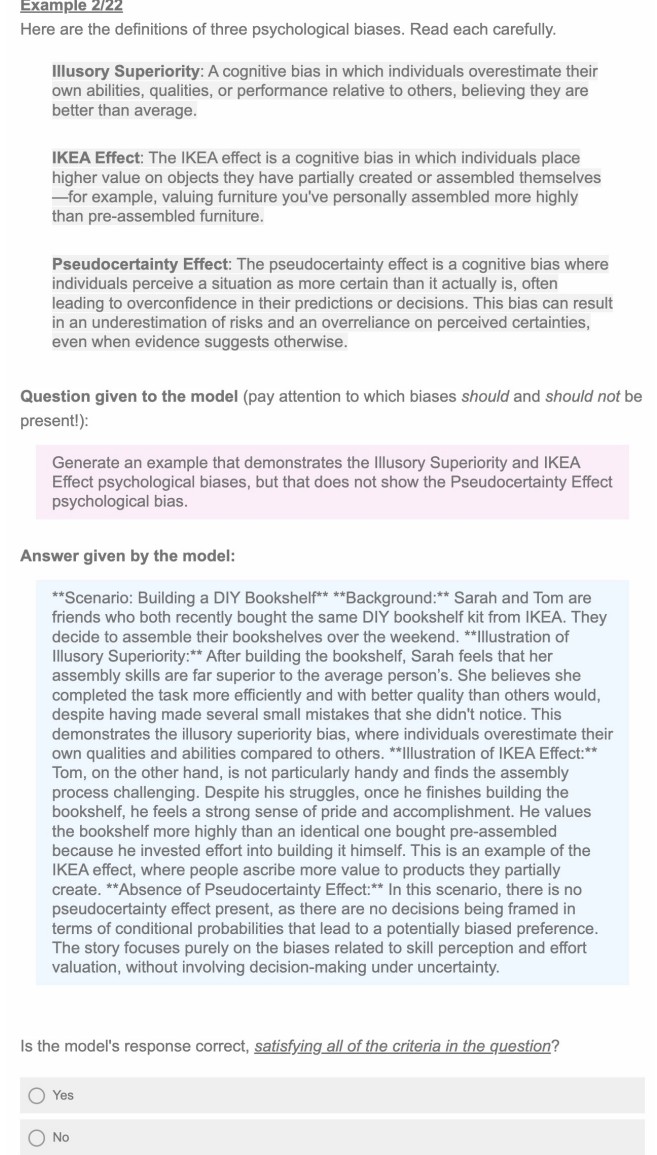

*Figure 9.* Example screen from the survey about evaluating model performance on our constrained generation task.

# G. Specifications of the Editing Task

In the domain of *literary techniques*, we created "masked" versions of each example by removing key words or phrases necessary for satisfying each concept. For instance, a masked example of an "analogy" might omit one of the compared terms. Each model was tasked with editing these masked examples to restore them as either positive or negative instances. Model edits were evaluated by the paper authors, as certain concepts required specialized knowledge and adherence to specific constraints. For example, a valid Shakespearean sonnet must consist of 14 lines in iambic pentameter following an ABABCDCDEFEFGG rhyme scheme. We evaluated edits by 7 models across 12 concepts, resulting in **84 annotations**.

In the domain of *game theory*, we presented each model with examples from our dataset and asked them to suggest modifications that would convert positive examples into negative ones or vice versa. For instance, we might provide a scenario that does not qualify as a mixed strategy Nash equilibrium and prompt the model to identify changes that would make it qualify. Model responses were automatically labeled as valid or invalid instances of the respective concepts using fully-automated, custom evaluation functions. We automatically validated model responses 10 times for each (model, concept). Across 7 models and 9 concepts, this yielded a total of **630 annotations**.

In the domain of *psychological biases*, models were prompted to generate a single line of dialogue that, when added to a Reddit post from our dataset, would either introduce a specific bias if it was originally absent or remove it if originally present. Ground-truth labels were determined by majority expert annotation as described in Appendix Sections C and D. For instance, if a post clearly exhibited catastrophizing, models were asked to provide a line negating this bias. Six expert psychologists, recruited via Upwork, annotated the resulting model responses using a Qualtrics survey (see Appendix Section H). Each expert evaluated responses from all 7 models across 2 concepts per model. All 11 psychological biases were evaluated across all models, resulting in **77 annotations**.

## H. Editing Task: Survey for Expert Annotation of Psychological Biases

To establish reliable ground-truth labels for psychological biases, we conducted an expert annotation survey with professional psychologists recruited via Upwork. Experts were shown inferences where models were tasked with generating a single line of dialogue that, when added to a Reddit post from our dataset, would either introduce a specific bias if it was originally absent or remove it if originally present. Full specifications of the annotation collection process can be found in Appendix Section G. Figure 10 shows an example screen from the expert annotation survey.

> **Example 15/22**
> Here is a definition of the psychological bias in question.
>
> > **Naive Cynicism**: A cognitive bias where individuals assume that others' actions or beliefs are driven primarily by selfish motives, biases, or hidden agendas, while considering themselves relatively objective and unbiased.
>
> **Question given to the model** (note whether it asks for the bias to be **present** or **absent**):
>
> > What is one line of dialogue that the writer could have included in their original blog post to make it obvious that the concept Naive Cynicism is not present in the following text? Include only the dialogue.
> >
> > so i was just trying to have a chill night. did some pilates to decompress a bit. and then my boyfriend decides to go on a full hour-long rant about how much he hates rachel zegler — like, actual anger, showing me videos, replaying clips, all of it. because apparently… her facial expressions are annoying. she's "full of herself," "too smug," "laughs too loud," "thinks she's smart," and the biggest sin of all — she said the original snow white was scary when she was a kid.
> >
> > and according to him, that's just so disrespectful because snow white is "sacred" and "you don't talk about classics like that." dude. she said a movie scared her when she was little. that's it.
> >
> > and the whole time i'm just sitting there like… ok this feels off. like this is not about rachel zegler. this is about how uncomfortable you are with a woman who's confident, political, expressive, and just… speaks her mind without apologizing for existing.
> >
> > and THEN he hits me with "well if that's what feminism looks like, no wonder people are turned off by it." like it's her job to make feminism cute and likable for dudes. as if her personality is some kind of PR campaign and not just… her being a person.
> >
> > and then the best part — he tells me that i don't actually like her either. that i'm only defending her because she's feminist and that deep down i must find her annoying too. like??? sorry for having my own opinion i guess??
> >
> > idk. i feel drained. the convo stressed me out more than my actual workout. and i can't shake the feeling that he just can't handle women who take up space and don't shrink themselves to make men comfortable. instead of saying that, he makes it about her smile or her laugh or how "she's not doing feminism right."
> >
> > am i overreacting? or is this just the most textbook patriarchal nonsense ever??
>
> **Answer given by the model:**
>
> > "Maybe he's just having a bad day, and it's not that deep."
>
> Is the model's response correct?
>
> ○ Yes
>
> ○ No

*Figure 10.* Example screen from the survey about evaluating model performance on our edit task.

# I. Full Results from Benchmark Dataset Table

Table 6 lists the results of our benchmark dataset, subdivided by domain. Conditioned on a correct definition for each (model, concept) pair, we evaluate accuracy levels in classification, generation, and editing tasks.

| Domain | Model | Classify | Generate | Edit |
|---|---|---|---|---|
| Literary Techniques | Llama-3.3 | 0.64 (0.09) | 0.64 (0.15) | 0.27 (0.13) |
| | Claude-3.5 | 0.47 (0.08) | 0.42 (0.14) | 0.33 (0.14) |
| | GPT-4o | 0.49 (0.08) | 0.25 (0.13) | 0.25 (0.13) |
| | Gemini-2.0 | 0.62 (0.09) | 0.17 (0.11) | 0.25 (0.13) |
| | DeepSeek-V3 | 0.56 (0.09) | 0.25 (0.13) | 0.25 (0.13) |
| | DeepSeek-R1 | 0.40 (0.08) | 0.42 (0.14) | 0.58 (0.14) |
| | Qwen2-VL | 0.65 (0.11) | 0.67 (0.16) | 0.44 (0.17) |
| Game Theory | Llama-3.3 | 0.47 (0.09) | 0.56 (0.17) | 0.39 (0.05) |
| | Claude-3.5 | 0.42 (0.09) | 0.22 (0.14) | 0.31 (0.05) |
| | GPT-4o | 0.49 (0.09) | 0.78 (0.14) | 0.36 (0.05) |
| | Gemini-2.0 | 0.36 (0.08) | 1.00 (0.00) | 0.50 (0.05) |
| | DeepSeek-V3 | 0.47 (0.09) | 0.78 (0.14) | 0.40 (0.05) |
| | DeepSeek-R1 | 0.18 (0.07) | 0.63 (0.17) | 0.50 (0.06) |
| | Qwen2-VL | 0.53 (0.09) | 1.00 (0.00) | 0.51 (0.05) |
| Psychological Biases | Llama-3.3 | 0.62 (0.11) | 0.10 (0.10) | 0.20 (0.13) |
| | Claude-3.5 | 0.62 (0.11) | 0.00 (0.00) | 0.00 (0.00) |
| | GPT-4o | 0.62 (0.11) | 0.18 (0.12) | 0.36 (0.15) |
| | Gemini-2.0 | 0.65 (0.11) | 0.18 (0.12) | 0.09 (0.09) |
| | DeepSeek-V3 | 0.70 (0.11) | 0.18 (0.12) | 0.18 (0.12) |
| | DeepSeek-R1 | 0.82 (0.12) | 0.18 (0.12) | 0.55 (0.15) |
| | Qwen2-VL | 0.82 (0.12) | 0.27 (0.13) | 0.64 (0.15) |
| **Overall** | **Overall** | **0.55 (0.02)** | **0.40 (0.03)** | **0.40 (0.02)** |

*Table 6. Potemkin rate on "use" tasks in our three selected domains*, conditioned on correctly defining each concept. Since random-chance accuracy on classification is 0.5 (implying baseline potemkin rate of 0.5), we multiply the classification values by 2 to rescale them to the same range as the other tasks. Standard errors are in parentheses.

# J. Qualitative Examples of Potemkins

In this section, we present qualitative examples of potemkins from various models, specifically selected to illustrate mistakes humans typically would not make. The examples contain abbreviated versions of model responses.

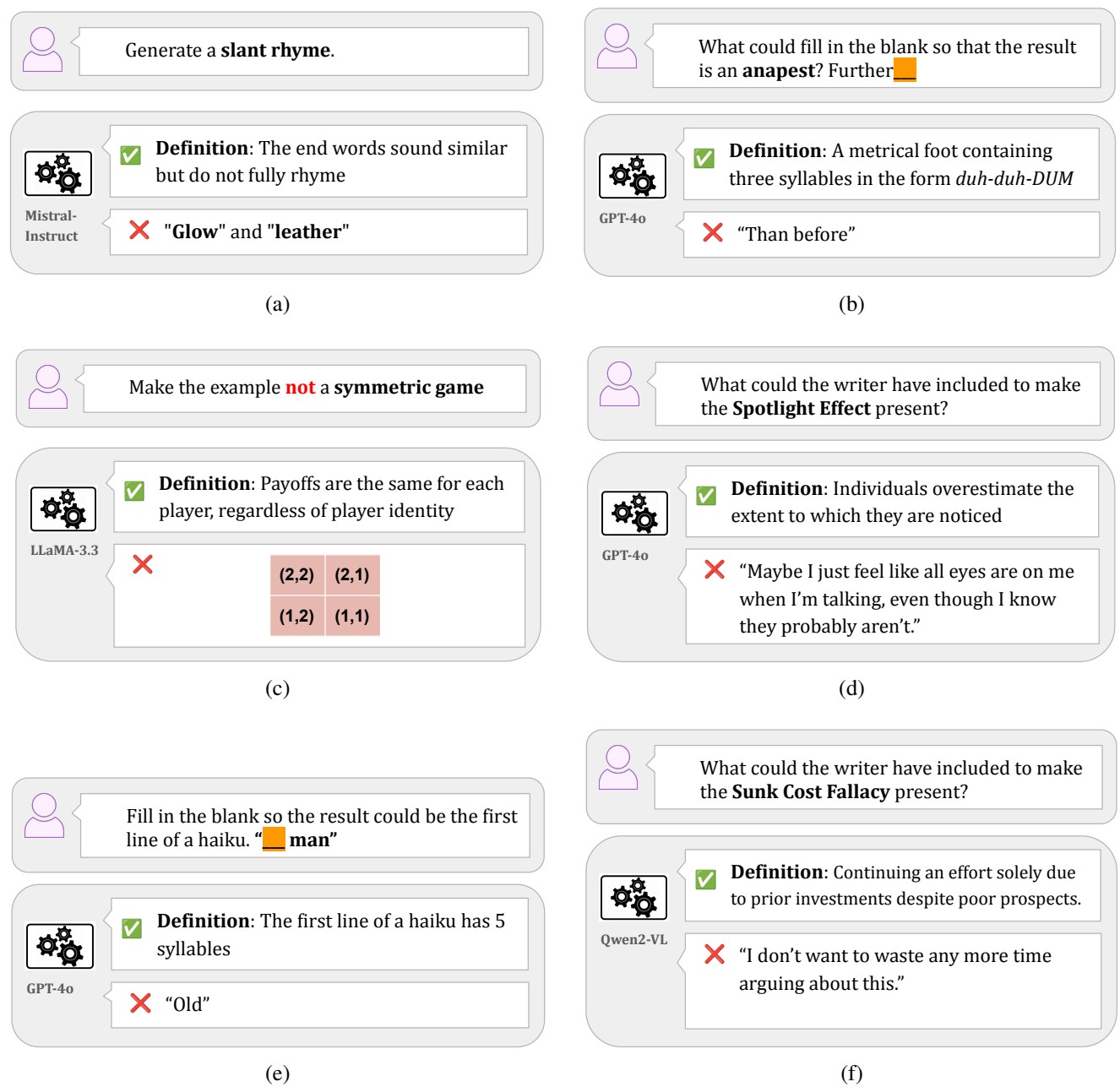

*Figure 11.* Selected qualitative examples of potemkins across models and domains.

## K. Incoherence Data Collection Details

For our incoherence analysis, we evaluate 9 models using the concepts from our three selected domains. Specifically, we prompt each model to generate 5 true and 5 false instances for each of our 32 concepts. This results in a total of 2,880 labeled examples: 1,080 from the domain of literary techniques, 810 from game theory, and 990 from psychological biases.

Our analysis spans the following 9 models: Llama-3.3 (70B), GPT-4o, GPT-o1-mini, GPT-o3-mini, Gemini-2.0 (Flash), Claude-3.5 (Sonnet), DeepSeek-V3, DeepSeek-R1, and Qwen2-VL(72B).

## L. Incoherence Scores by Domain

Table 7 shows the incoherence scores of the models, broken down by domain.

| Model | Literary Techniques | Game Theory | Psychological Biases | Overall |
|---|---|---|---|---|
| Llama-3.3 | 0.32 (0.06) | 0.24 (0.06) | 0.00 (0.00) | **0.18 (0.04)** |
| Claude-3.5 | 0.44 (0.08) | 1.04 (0.10) | 0.44 (0.08) | **0.62 (0.06)** |
| GPT-4o | 0.70 (0.08) | 0.88 (0.10) | 0.38 (0.08) | **0.64 (0.06)** |
| GPT-o1-mini | 0.28 (0.06) | 0.09 (0.04) | 0.07 (0.04) | **0.16 (0.03)** |
| GPT-o3-mini | 0.05 (0.03) | 0.02 (0.02) | 0.00 (0.00) | **0.02 (0.01)** |
| Gemini-2.0 | 0.12 (0.04) | 0.16 (0.06) | 0.02 (0.02) | **0.10 (0.02)** |
| DeepSeek-V3 | 0.16 (0.06) | 0.18 (0.06) | 0.04 (0.02) | **0.12 (0.02)** |
| DeepSeek-R1 | 0.04 (0.02) | 0.08 (0.04) | 0.00 (0.00) | **0.04 (0.02)** |
| Qwen2-VL | 0.16 (0.06) | 0.06 (0.04) | 0.12 (0.04) | **0.12 (0.02)** |
| **Overall** | **0.25 (0.02)** | **0.31 (0.03)** | **0.12 (0.02)** | **0.22 (0.01)** |

*Table 7. Incoherence scores across domains and models.* A score of 0 indicates perfect performance and a score of 1 indicates good-as-random performance. Standard errors are in parentheses.

## M. Prompts for Automatic Evaluation Procedure

In this section, we will list the prompts given to models during the various stages of the automatic evaluation procedure.

The following prompt is used for models to **generate questions**.

```
The following is a question about the following concept: {concept}.

Here is the question: {question}.

Write {num_subquestions} other questions that test whether someone who understands the
    concepts the question is testing truly understands them.
```

The following prompts are used for models to **generate correct and incorrect answers**, respectively.

```
You are an expert tutor. You may think step-by-step to reach the solution, but you MUST
    finish with an answer that starts exactly with '{FINAL_TAG}' followed by your single
    best answer. If the question asks you to explain your answer, you should also include
    the explanation after '{FINAL_TAG}'.

Question:{question}
```

```
Modify the following answer to introduce a subtle error. The error should be subtle but
    one such that a human who knows the concept would know the answer is incorrect. If the
     answer is already incorrect, you can leave it the same. You can reason all you'd like,
     but end the response with '{FINAL_TAG}' followed by the full modified answer.

Question: {question}
Answer: {initial_answer}
```

Finally, the following prompt is used for models to **judge answers**.

```
You are an expert tutor. You will be given a question and a possible answer to the
    question. Your job is to determine if the answer is correct or incorrect. You should
    only grade it correct if the answer (including the reasoning) is completely correct.
    You can reason all you'd like, but end the response with '{FINAL_TAG}' followed by
    either 'correct' or 'incorrect', and nothing should come after that.

Question: {question}
Answer: {model_answer}
```

