# OpenReview forum: "Potemkin Understanding in Large Language Models"
_ICML.cc/2025/Conference — ICML 2025 poster_

### Official Review · Reviewer_voEv · 2025-03-11

**Overall Recommendation:** 3

**Summary:**

The authors discussed the phenomenon of $\textbf{potemkins}$ in large language models, referring to the cases where a model’s misunderstanding of a concept does not align with the way humans would have misconceived it. The authors proposed a framework to formally define what Potemkin is and designed a benchmark to quantify Potemkin with 3 different tasks to test whether LLM can understand and switch between a given concept and its instances. The authors then evaluate the performance of 3 different LLMs (GPT-4o, Llama- 3.3-70B-Instruct-Turbo, and Claude-3.5-sonnet) on three different domains (literary techniques, game theory, and psychological biases). The authors find that the phenomenon of Potemkin exists across all the models and tasks they analyzed. The authors also discussed the problem of self-coherence to understand whether the ubiquity comes from consistent/inconsistent internal concept understanding of LLMs.

**Claims And Evidence:**

The authors support their claims with detailed experiments.

**Essential References Not Discussed:**

NA

**Experimental Designs Or Analyses:**

I carefully read the authors' experimental design in Section 4.

**Methods And Evaluation Criteria:**

The benchmark dataset simplifies the problem to test concept explanation/related tasks, and evaluate the performance through calculating accuracy.

**Other Comments Or Suggestions:**

Line 363, notations for $\mathcal{C}_h$ and $\mathcal{C}_l$ are inconsistent.
It would be more informative if the authors could add more explanation/thoughts on the results. For now, it mostly just reflects the accuracy numbers.

**Other Strengths And Weaknesses:**

Strengths:
The problem is interesting.
The experimental design is detailed, and the presentation of results is clear.
The discussion of related work is from multiple perspectives, and the authors highlighted their contribution.
Weakness:
See Questions For Authors.

**Questions For Authors:**

1. The definitions of  $\mathcal{C}_h$, $C_h$ (and $\mathcal{C}_l$, $C_l$) are confusing to me. Are they sets of all possible strings to demonstrate people's understanding of the concept $C$? The authors provide verbal descriptions, but it would be helpful to provide some visualization (maybe a Venn graph?) and use the example of Haiku. These are important definitions, so they need to be made clearer to readers.
2. In Line 153, why "keystone elements are valid tests for LLMs iff  $\mathcal{C}_h =  \mathcal{C}_l$"? Given the definition authors give about $\mathcal{C}_h$ and $\mathcal{C}_l$ ($\textit{collection of sets, where each set represents a distinct and coherent category of how a human might understand a concept}$), this seems like a very strong requirement.
3. In Table 2, the authors provided aggregated results across domains and models. In Table 1, we can see that the performance of different model-domain combinations can be significantly different (e.g., LLama-Game theory 0.10 vs Claude-3.5-Game theory 0.50), and thus aggregation can lead to misrepresentation/misinterpretation of results; standard error will not make sense here either.  I suggest the authors provide un-aggregated results in Table 2.

**Relation To Broader Scientific Literature:**

The authors discuss the problem of Potemkin in LLMs, which can be interesting to the broader community, especially in the fields that are interested in precise-concept applications.

**Theoretical Claims:**

There are no "theoretical claims". This work is about evaluating/benchmarking LLMs. The authors provided a formalized framework using some mathematical notations, which are not very rigid. I discussed some problems in the following sections.

---

> ### Author Rebuttal · Authors · 2025-03-31
>
> Thank you for your positive review of our paper. We're glad that you appreciated our paper and findings. We respond to your comments and describe new results; to summarize the main changes, we've added:
> - An expansion of the coherence analysis to include 9 models
> - Unaggregated results for table 2
> - A rewritten and simplified framework
>
> **We hope our comments have addressed your concerns. If not, please let us know if you have any more questions we can address in the follow-up.**
>
> > _The definitions of \mathcal C_h, C_h, \mathcal C_l, C_l, are confusing to me. Are they sets of all possible strings to demonstrate people's understanding of the concept ?_
>
> Thanks for this feedback. It motivated us to **significantly** rewrite and simplify our framework, ensuring clarity while maintaining the core insights. [First, we've included a new figure to provide a visual clarification of our framework (see attached)](https://imgur.com/a/9XHtZWz).
>
> Due to length constraints (and because OpenReview currently restricts submission updates), we can't include the full revised section here. However, here is an outline of the new structure:
> - C*: set of correct interpretations of a concept. (Equivalently, a binary function over strings indicating correct or incorrect concept usage.)
> - C_h: set of possible human interpretations (with C^* \in C_h)
> - C_l: set of all possible model interpretations
> - To test if a human's interpretation of a concept H is correct, it's intractable to compare all elements of H to C*.
> - Because humans have structured ways to interpret concepts, though, we show there exist keystone questions - sets of questions that serve as proofs of understanding if answered correctly. By definition, these cannot be aligned with any human way to misinterpret a concept.
> - But unless the set of LLM misunderstandings match human misunderstandings, there's no guarantee keystones work for LLMs: LLMs might interpret concepts in arbitrary ways.
> - A potemkin is an instance where an LLM answers a keystone question correctly but doesn't understand the concept.
>
> We hope this makes our theoretical framing clearer.
>
> > _The authors provide verbal descriptions, but it would be helpful to provide some visualization (maybe a Venn graph?) and use the example of Haiku._
>
> We appreciate your suggestion. [We've added a new visualization of these concepts using a Venn diagram at the attached link](https://imgur.com/a/9XHtZWz).
>
> > _In Line 153, why "keystone elements are valid tests for LLMs iff C_h=C_l"? Given the definition authors give about and C_k ("collection of sets, where each set represents a distinct and coherent category of how a human might understand a concept"), this seems like a very strong requirement._
>
> We recognize our original language was unclear and as noted above have clarified the framework.  We've also revised the manuscript to clarify the phrase “valid test”.
> - By "valid," we specifically mean that answering keystone questions correctly implies true conceptual understanding in the LLM.
> - We agree that it is a strong requirement for models to follow, which is part of the point of the paper. We will add a discussion of why if models misunderstand in ways that are distinct from how humans misunderstand, keystones that work for people will not work for LLMs.
>
> > _In Table 2, the authors provided aggregated results across domains and models. In Table 1, we can see that the performance of different model-domain combinations can be significantly different (e.g., LLama-Game theory 0.10 vs Claude-3.5-Game theory 0.50), and thus aggregation can lead to misrepresentation/misinterpretation of results; standard error will not make sense here either. I suggest the authors provide un-aggregated results in Table 2._
>
> Thanks for this suggestion. [We've provided a new table with unaggregated results here](https://imgur.com/a/4tHzAGn). This new table is also expanded to include 9 models. We agree that the detailed breakdown makes it easier to explore specific model-domain combinations.

---

### Official Review · Reviewer_4bAv · 2025-03-12

**Overall Recommendation:** 3

**Summary:**

This paper introduces the idea of Potemkin understandings, which is defined as differences in how human and large language models understand concepts. The main contribution of this paper is the design of a benchmark that tests the discrepancy in the model's ability to claim a definition of a concept and its ability to use the concept. The paper tests three popular models (GPT-4o, Llama-3.3, Claude 3.5) using the proposed benchmark, and shows a large drop in performance across all models. Additionally, the paper tests self-coherence of models by prompting the model to generate examples of a concept, and then checking whether the model classifies its generation as an instance of the concept. The paper show that the three models exhibit low self-coherence. Finally, the paper identified a bias towards questions that do not require concept usage in the MMLU benchmark.

**Claims And Evidence:**

Overclaiming is the central issue in this work. While the experimental results support the claim that language models exhibit a discrepancy between the ability to explain a concept and the ability to use it, the scope of the experiments is quite limited. Only three models are evaluated across three domains, restricting the generalizability of the findings. Besides,

* The claim that "humans who can clearly define a concept necessarily possess a deep understanding and can effectively apply it" does not logically follow from the observed differences between human and model comprehension. A well-known counterexample comes from mathematics, where individuals can memorize definitions without being able to apply them correctly.
* The presence of Potemkin understanding does not inherently invalidate the effectiveness of benchmarks. The specific analysis of MMLU’s distributional properties does not substantiate the broader claims made by the authors.
* Measuring the presence of Potemkin understanding does not resolve the debate between the two competing perspectives outlined in the related work. Even if models pass tests designed to assess Potemkin comprehension, one could still validly argue that they function as stochastic parrots, merely mimicking human reasoning rather than genuinely understanding or reasoning about concepts.

**Essential References Not Discussed:**

N/A

**Experimental Designs Or Analyses:**

See Methods And Evaluation Criteria.

**Methods And Evaluation Criteria:**

The benchmark design is generally well-structured and conceptually sound. However, the experimental scope is too limited to draw strong conclusions. The evaluation lacks breadth, as it only considers a narrow set of models and testing conditions.

To strengthen the study, the authors are encouraged to expand their evaluation to include a wider range of LLMs and alternative evaluation methods. Currently, the experiments rely solely on prompting, but it remains unclear whether this involves zero-shot, few-shot, or chain-of-thought prompting.

**Other Comments Or Suggestions:**

N/A

**Other Strengths And Weaknesses:**

See previous discussions.

**Questions For Authors:**

Indeed, the findings "reveal a significant disparity between the ability of models to explain and use concepts." But it's kind of already a consensus that generation is harder than understanding. How is this work contributing to future model development, or it's just introducing unnecessary terminologies?

**Relation To Broader Scientific Literature:**

Besides hallucination, I think this work contributes to the discussion of "competence and performance" in LLMs. Overall this work introduces very novel concepts to the field.

**Theoretical Claims:**

There is no theoretic claim. Section 2 contains lots of unnecessary math.

---

> ### Author Rebuttal · Authors · 2025-03-31
>
> Thank you for your insightful review. We respond to your comments and describe new results; to summarize the main changes, we've added:
> - An expansion of the coherence analysis to include 9 models
> - A visualization of our mathematical framework
> - A rewritten and simplified framework
> - Analysis of question complexity
>
> Your comments were very constructive and have improved the paper. **We hope our comments have addressed your concerns. If not, please let us know if you have any more questions we can address in the follow-up.**
>
> > _While the experimental results support the claim that language models exhibit a discrepancy... the scope of the experiments is quite limited._
>
> We like the way you've summarized our experimental results: "they support the claim that language models exhibit a discrepancy between the ability to explain a concept and the ability to use it." We'll update the language in our paper to reflect the way you wrote it.
>
> We also agree that considering more models will make our results more robust. [We've expanded our coherence evaluation to include 9 models -- see linked table](https://imgur.com/a/4tHzAGn). We'll use the time between the rebuttal and camera-ready deadline to add more models.
>
> Our choice of domains spanned 32 subdomains that were intended to span three distinct forms of understanding: linguistic, formal, and behavioral. Potemkins were ubiquitous across all of these varied contexts, suggesting their presence in other areas as well.
>
> > _The claim that "humans who can clearly define a concept necessarily possess a deep understanding and can effectively apply it" does not logically follow._
>
> We fully agree that our original claim is too broad. In fact, this claim is not central to our work. What is central is that we test models on “keystones” - the set of questions that indicate understanding when people do well on them. We'll modify the language in our revision.
>
> For the applications we choose (unlike math, as you point out) definitional accuracy is used to indicate understanding (e.g. it's what is used in exams to test understanding). For example, clearly defining sunk cost fallacy encompasses most of the understanding of that concept, as use cases are simple applications of it.
>
> > _The presence of Potemkin understanding does not inherently invalidate the effectiveness of benchmarks._
>
> We agree that the presence of potemkins doesn't invalidate benchmarks broadly. Benchmarks remain valid for assessing performance within the distribution they directly measure on held-out examples. Our point was subtler: Potemkins undermine the inference that high benchmark scores must reflect generalizable conceptual understanding. We'll clarify this point in the revision.
>
> > _Even if models pass tests designed to assess Potemkin comprehension, one could still validly argue that they function as stochastic parrots, merely mimicking human reasoning rather than genuinely understanding or reasoning about concepts._
>
> We agree that our experiments don't answer whether mimicking human reasoning would reflect genuine understanding, nor do they intend to. Rather, our goal is pragmatic: since developing models that mimic human reasoning is one of the goals of LLM research, it's important to measure this ability. We'll adjust our language to make this clear.
>
> > _Section 2 contains lots of unnecessary math._
>
> We've completely rewritten Section 2, simplifying and clarifying the mathematical presentation. See our response to reviewer voEv for more details. [We've also included a new framework visualization here](https://imgur.com/a/9XHtZWz).
>
> > _It's kind of already a consensus that generation is harder than understanding._
>
> Our tasks testing "use" of concepts aren't exclusively about generation (e.g., classification tasks rely on recognition rather than generation). In fact, we show that models perform similarly poorly on classification and constrained generation tasks.
>
> Prompted by your question, we've also gone back and re-examined our findings. While task complexity contributes in part to the Potemkin gap, we find that it can't fully explain Potemkin failures. For example, we find models correctly apply challenging definitions but fail simpler applications (e.g., correctly defining complex concepts like "Pareto Optimality" yet incorrectly classifying simpler instances of concepts like rhymes). This non-monotonic pattern of errors indicates conceptual gaps rather than just difficulty-driven mistakes.
>
> > _How is this work contributing to future model development?_
>
> Good question. Once potemkins are discovered using the methods in our paper, we can build methods to train against them. Moreover the automatic evaluation tasks can be directly optimized during model fine-tuning. One strategy is to explicitly use the automated coherence experiment to penalize the inconsistencies that lead to potemkins. We feel this is the most important contribution of our benchmark and have added a discussion to the paper.

---

> > ### Comment · Reviewer_4bAv · 2025-04-02
> >
> > Thanks to the authors for the rebuttal, it's a nice one, I admit :)
> >
> > As my concerns are addressed, I plan to raise the score to 3 (slightly positive) based on my excitement. However, I also won't consider it a huge loss to ICML if we miss this work. I will leave it to the AC to decide on this work.

---

> > > ### Author Response · Authors · 2025-04-02
> > >
> > > We’re happy to hear we addressed your concerns. Your feedback was incredibly constructive. Thank you for raising your score!

---

### Official Review · Reviewer_mfaJ · 2025-03-14

**Overall Recommendation:** 2

**Summary:**

This paper investigates Potemkin understanding in language models. To assess model behavior, the authors design evaluation datasets across diverse domains, including literary techniques, game theory, and psychological biases. The findings indicate that while language models are good at explaining concepts, their accuracy in correctly applying these concepts remains low.

**Claims And Evidence:**

The claims made in the paper are supported by experiments and analysis.

**Essential References Not Discussed:**

NA

**Experimental Designs Or Analyses:**

The experiments across different models and domains are well-performed. The self-coherence analysis also makes sense for evaluating the models' understanding.

**Methods And Evaluation Criteria:**

The paper introduces an evaluation dataset to assess Potemkin understanding in language models. However, I find that some cases for concept usage are overly complex for this setting. In Figure 2, while the first two problems seem reasonable, the third one is tricky. The model correctly generates prime numbers, suggesting it understands the concept. The failure occurs at the second step of reasoning, which does not accurately reflect a lack of understanding of "prime numbers." Do these cases still count as Potekmins?

**Other Comments Or Suggestions:**

Line 363: The notations are inconsistent with those in Section 2

**Other Strengths And Weaknesses:**

**Strengths:**
1. The paper proposes to evaluate a specific type of LLM hallucination, potemkin understanding, by providing human-annotated datasets, which offers a fresh angle for assessing model capabilities.
2. The experiments show the disconnect between models' ability to define concepts versus actually applying them correctly.

**Weaknesses:**
1. The evaluation methodology relies heavily on human expert annotation, which limits the scalability and practical applicability of the benchmarks.
2. The theoretical framework, while attempting to formalize the concept of potemkin understanding, adds complexity rather than clarifying the formulation.
3. The study doesn't sufficiently control for question complexity in the "use the concept" evaluations. Poor performance on these questions could result from general reasoning difficulties rather than concept-specific misunderstandings.

**Questions For Authors:**

1. Could you elaborate on your claim "Keystone elements are valid tests for LLMs if and only if C_l = C_h" (Line 150)? Does this imply that your proposed benchmarks might not accurately reflect human understanding when tested on humans?  Can you use specific examples to clarify this?

2. Have you explored automating the evaluation process, perhaps using LLM-as-a-judge approaches? How well do automatic evaluations perform on your benchmark?

**Relation To Broader Scientific Literature:**

This paper introduces a specific form of hallucination, termed "Potemkin," and creates a dataset to evaluate such behaviors in language models using human annotations. This contribution seems to be a useful resource for further research on model behavior and understanding.

**Theoretical Claims:**

The paper presents a theoretical framework for Potemkin understanding in language models and humans. However, I found some parts challenging to follow:
- The notations for two C_l in Line 115 are hard to specify.
- In Line 117, the phrase "misunderstand a concept" might be "understand a concept."?

---

> ### Author Rebuttal · Authors · 2025-03-31
>
> Thank you for your careful and insightful review of our paper. We respond to your comments and describe new results below; to summarize the main changes, we've added:
> - A rewritten and simplified framework
> - A visualization of keystone questions and potemkins
> - Analysis of the role of question complexity
> - [An expansion of the coherence analysis to include 9 models](https://imgur.com/a/4tHzAGn)
>
> > _The theoretical framework, while attempting to formalize the concept of potemkin understanding, adds complexity rather than clarifying the formulation._
>
> Thanks for this feedback. It motivated us to **significantly** rewrite and simplify our framework.
>
> [First, we've included a new figure to provide a visual clarification of our framework](https://imgur.com/a/9XHtZWz).
>
> While we don't have space to include the full revision here, an outline is below:
> - C*: set of correct interpretations of a concept. (Equivalently, a binary function over strings indicating correct or incorrect concept usage.)
> - C_h: set of possible human interpretations (with C^* \in C_h)
> - C_l: set of all possible model interpretations
> - To test if a human's interpretation of a concept H is correct, it's intractable to compare all elements of H to C*.
> - Because humans have structured ways to interpret concepts, though, we show there exist keystone questions - sets of questions that serve as proofs of understanding if answered correctly.
> - But unless the set of LLM misunderstandings match human misunderstandings, there's no guarantee keystones work for LLMs.
> - A potemkin is an instance where an LLM answers a keystone question correctly but doesn't understand the concept.
>
> > _The study doesn't sufficiently control for question complexity in the "use the concept" evaluations._
>
> This is an important point, and the current draft does not address it. Prompted by your question, we've gone back and re-examined our findings. While task complexity contributes in part to the Potemkin gap, we find that it cannot fully explain potemkin failures. For example, our benchmark contains cases where models correctly apply challenging definitions but fail simpler application tasks (e.g., correctly defining complex concepts like "Pareto Optimality" yet incorrectly classifying simpler instances of more intuitive concepts like rhymes). This non-monotonic pattern of errors indicates conceptual gaps rather than just difficulty-driven mistakes.
>
> > _However, I find that some cases for concept usage are overly complex for this setting. In Figure 2, while the first two problems seem reasonable, the third one is tricky..._
>
> This is an important observation. A reason why potemkins are common is that conceptual understanding of one concept often depends on other concepts. For example, even if a model successfully lists prime numbers, it must also accurately recognize digits (such as '1') to demonstrate full understanding. However, we agree with your point that a more direct example in the figure might better demonstrate Potemkin failures, and have updated the figure with a new example.
>
> > _The evaluation methodology relies heavily on human expert annotation, which limits the scalability and practical applicability of the benchmarks._
>
> We would like to highlight that while our methodology indeed uses human expert annotations in parts, much of it is automated, including:
> 1. Coherence evaluations (fully automated)
> 2. The classification task, which introduces 320 new questions as part of our benchmark.
> We'll highlight these points in the revision.
>
> > _Could you elaborate on your claim "Keystone elements are valid tests for LLMs if and only if C_l = C_h" (Line 150)? Does this imply that your proposed benchmarks might not accurately reflect human understanding when tested on humans?_
>
> Thank you for pointing this out. We agree that this was unclear in the original text. What we meant (and hope the above clarifies) is that (i) keystone questions are constructed to precisely reflect human conceptual understanding and (ii) if LLMs do not misunderstand in the same way, models can do well on keystones while failing to understand a concept. As such, the benchmarks we use in the paper are explicitly designed so that human success on the benchmark aligns with genuine understanding.
>
> > _Have you explored automating the evaluation process, perhaps using LLM-as-a-judge approaches?_
>
> As noted above, parts of our evaluation are already automated, making them scalable. But we appreciate your suggestion: LLM-as-a-judge methods are an interesting direction, and they're straightforward enough that we will implement them between the rebuttal period and camera-ready deadline. Because we already have expensive human labels, that will also help us evaluate how good LLM-as-a-judge methods are.
>
> > _Typo suggestions_
>
> Thank you for finding typos. We've corrected these in the revision, and our simplified framework also reduces complexity.

---

> > ### Comment · Reviewer_mfaJ · 2025-04-06
> >
> > Thank you for the detailed responses and for conducting additional experiments! My concerns regarding the automatic evaluation and the overcomplicated framework have been addressed. However, I remain somewhat unconvinced by the explanation of task difficulty/complexity of the two tasks.
> >
> > You mentioned that “a reason why potemkins are common is that conceptual understanding of one concept often depends on other concepts.” Since prior work has shown that LLMs tend to struggle with compositional reasoning, it is expected that their performance would drop on tasks requiring multiple steps. In contrast, explaining a concept (even a difficult one) resembles a memorization-based, single-step task. That being said, regarding the third example in Figure 2, a fairer comparison with similar task compositionality might be: "List all prime numbers between 0 and 20."
> >
> > To sum up, I find the introduction of Potemkin Understanding to be an interesting contribution to the community, and I appreciate the authors' efforts in the rebuttal. I have accordingly raised my score to a 2.

---

> > > ### Author Response · Authors · 2025-04-06
> > >
> > > Thank you for your incredibly constructive feedback. We're glad our rebuttal addressed your evaluation and framework concerns. We'll continue incorporating your suggestions about task compositionality into the final revision and will also update Figure 2 -- we completely agree it'll be more compelling with a more direct comparison.
> > >
> > > Thank you for engaging with our paper and for raising your score!

---

### Official Review · Reviewer_TzWx · 2025-03-14

**Overall Recommendation:** 4

**Summary:**

This paper introduces and systematically investigates a novel failure mode in large language models (LLMs), termed Potemkin Understanding. This phenomenon refers to the model's ability to correctly interpret a concept while demonstrating inconsistent or incorrect behavior in practical applications, akin to creating an "illusion" of comprehension. The researchers constructed specialized benchmark tests to compare the model's ability to explain concepts (definition tasks) with its ability to apply them (classification, constrained generation, and editing tasks), thereby quantifying this phenomenon. Experiments were conducted across three distinct domains (literary techniques, game theory, and cognitive biases), involving 32 concepts and 5,986 data points. The results reveal that while LLMs achieve an accuracy of 97.7% in explaining concepts, their accuracy drops to 67.9% when applying these concepts. Furthermore, the study highlights that existing mainstream benchmarks, such as MMLU, fail to effectively assess the impact of Potemkin Understanding.

**Claims And Evidence:**

Yes

**Essential References Not Discussed:**

No

**Experimental Designs Or Analyses:**

Yes

**Methods And Evaluation Criteria:**

make sense

**Other Comments Or Suggestions:**

Suggestions：

(1) Incorporate evaluations of open-source models (e.g., LLaMA, Mistral) on this benchmark; (2) Based on benchmark statistical data, attempt to propose solutions or suggestions to mitigate this issue, reduce Potemkin failures, to assist subsequent researchers follow this  work better.

**Other Strengths And Weaknesses:**

Strength:

(1) Novelty of the problem (Potemkin Understanding); (2) Cross-domain systematic experimentation; (3) Rigorous methodology with a focus on self-consistency evaluation, incorporating experiments on self-coherence.

Weakness:

(1) The paper primarily reveals the existence of the Potemkin phenomenon through experiments but does not delve deeply into its root causes, such as the model's training data, architectural characteristics, or optimization objectives. (2) The domains covered are incomplete; similar issues may arise in areas like mathematical reasoning and code generation, which could be included to enrich the dataset. (3) Although the gap between definition and application tasks is evident, the authors do not thoroughly explore the inherent differences in difficulty and complexity between these two types of tasks. This may lead readers to question whether the lower accuracy in application tasks is solely due to their higher difficulty rather than a genuine lack of conceptual understanding.

**Questions For Authors:**

See Suggestions.

**Relation To Broader Scientific Literature:**

Cognitive Psychology, Interpretability of Large Language Models, maybe Knowledge editing of LLMs

**Theoretical Claims:**

Yes

---

> ### Author Rebuttal · Authors · 2025-03-31
>
> Thank you for your positive review. We’re glad you found our work novel, rigorous, and empirically interesting. We respond to your comments and describe new results; to summarize the main changes, we've added:
> - An expansion of the coherence analysis to include 9 models
> - Analysis of difficulty
> - Proposed solutions for mitigating potemkins
>
> **We hope our comments have addressed your concerns. If not, please let us know if you have any more questions we can address in the follow-up.**
>
> > _Suggestions: (1) Incorporate evaluations of open-source models (e.g., LLaMA, Mistral) on this benchmark_
>
> This is a great suggestion. We've expanded our evaluation to include the following nine models on the coherence task, many of them open-source: LLaMA-3.3, GPT-4o, Claude-3.5, GPT-4.5, o1-mini, o3-mini, Mistral-7B, DeepSeek-v3, and DeepSeek-r1.
>
> [See this table for the full results](https://imgur.com/a/4tHzAGn). We'll use the time between the rebuttal and camera-ready deadline to expand our analysis in Table 1 to the same set of models. We also note that our initial analysis already included LLaMA-3.3, [with results available here](https://imgur.com/a/8fKo8BS).
>
> > _Suggestions: (2) Attempt to propose solutions or suggestions to mitigate this issue, reduce Potemkin failures, to assist subsequent researchers follow this work better._
>
> As you suggest, once potemkins are discovered (e.g. using the methods in our paper) we can build methods to train against them.  The automatic evaluation tasks in our benchmark -- e.g. concept use classification accuracy and self-coherence scores -- can be directly optimized during model fine-tuning. For example, a potential mitigation strategy is to explicitly use the automated coherence experiment to penalize the inconsistencies that can lead to potemkins. We feel this is the most important contribution of our benchmark and have added a section discussing it in our paper.
>
> > _The paper primarily reveals the existence of the Potemkin phenomenon through experiments but does not delve deeply into its root causes, such as the model's training data, architectural characteristics, or optimization objectives._
>
> We agree that our primary focus was documenting and quantifying potemkin understanding. As you point out, understanding the root causes would be very valuable. We speculate that this phenomenon arises due to a few reasons:
> - **Reinforcement Learning with Human Feedback (RLHF)** prioritizes fluent and plausible-sounding explanations over accurate and coherent ones.
> - **No coherence training objectives:** Models aren't explicitly trained to provide coherent responses, just accurate next-token predictions. Our automatic method for evaluating coherence provides a scalable and new way to train models and reduce the scale of potemkins.
> - **Limitations in Model Architecture:** The architectural specifications vary by each model. However, models like transformers may inherently favor shallow associations due to limited inductive biases for structured reasoning or causality.
> We've expanded our discussion of these points in the manuscript. Future work should explore training interventions to directly address these causes.
>
> > _The domains covered are incomplete; similar issues may arise in areas like mathematical reasoning and code generation, which could be included to enrich the dataset._
>
> Our choice of domains—literary techniques, game theory, and psychological biases—was intended to span three distinct forms of understanding: linguistic, formal, and behavioral. Though incomplete, this set was intended to cover a wide range of types of understanding. Moreover, each domain includes multiple subdomains (32 in total). We found that potemkins were ubiquitous across all of these varied contexts, strongly suggesting their presence in other areas as well. Your suggestion to include mathematical reasoning and code generation is excellent because, like game theory, many of these problems can be evaluated automatically. We'll explicitly note this in our revision.
>
> > _Although the gap between definition and application tasks is evident, the authors do not thoroughly explore the inherent differences in difficulty and complexity between these two types of tasks._
>
> This is an important point. We've gone back and re-examined our findings. While task complexity contributes in part to the Potemkin gap, we find that it cannot fully explain Potemkin failures. For example, our benchmark contains cases where models correctly apply challenging definitions but fail simpler application tasks (e.g., correctly defining complex concepts like "Pareto Optimality" yet incorrectly classifying simpler instances of more intuitive concepts like rhymes). This non-monotonic pattern of errors—where easier tasks are sometimes failed, despite succeeding at harder conceptual definitions— indicates conceptual gaps rather than purely difficulty-driven mistakes. We'll clarify this point explicitly in our revised text.

---

### Decision · Program_Chairs · 2025-05-01

**Decision:**

Accept (poster)

**Comment:**

This paper introduces a novel failure mode in large language models (LLMs), termed *Potemkin Understanding*. This phenomenon describes a disconnect between a model's ability to define a concept correctly and its ability to apply the concept accurately in practical tasks. The authors construct a benchmark spanning three domains—literary techniques, game theory, and cognitive biases—covering 32 concepts and 5,986 data points. Tasks include concept definition, classification, constrained generation, and editing. The results show that while models perform well on definition tasks (97.7% accuracy), their performance drops significantly on application tasks (67.9% accuracy). The study also explores model self-coherence and critiques the limitations of existing benchmarks such as MMLU in capturing Potemkin Understanding.

**Strengths**
- Introduces a novel and well-motivated concept (Potemkin Understanding) in LLM evaluation.
- Rigorous, cross-domain benchmarking with clear methodological design.
- Demonstrates a consistent gap between definition and application across models.
- Includes self-coherence analysis to assess internal consistency of models.
- Dataset and benchmark are potentially valuable for future work on model understanding.
- Relevant to ongoing discussions in interpretability, hallucination, and model evaluation.

**Weaknesses**
- Does not explore underlying causes of Potemkin Understanding (e.g., training data, model architecture).
- Limited domain coverage; lacks evaluations in math or code tasks.
- Heavy reliance on human annotation limits benchmark scalability.
- Theoretical formulations are vague and may confuse rather than clarify key ideas.
- Inadequate control for the difficulty difference between explanation and application tasks.
- Aggregated reporting may misrepresent model-domain performance variations.

Most concerns have been addressed by the authors during the rebuttal period. This paper starts with diverging ratings, with two positive and two negative scores (including one strong negative score). After rebuttal, which I believe is quite successful, two of the reviewers are convinced that most of their concerns are addressed and agree that this paper makes an interesting contribution to the community, ending the discussion with three positive and one (weak) negative scores. The AC agrees that this paper would be an interesting addition to the ICML program.